# Short-Periods of Pre-Warming in Laparoscopic Surgery. A Non-Randomized Clinical Trial Evaluating Current Clinical Practice

**DOI:** 10.3390/jcm10051047

**Published:** 2021-03-03

**Authors:** Ángel Becerra, Lucía Valencia, Jesús Villar, Aurelio Rodríguez-Pérez

**Affiliations:** 1Department of Anesthesiology, Hospital Universitario de Gran Canaria Doctor Negrín, 35010 Las Palmas de Gran Canaria, Spain; ori98es@yahoo.es (L.V.); arodperp@gobiernodecanarias.org (A.R.-P.); 2Department of Medical and Surgical Sciences, Universidad de Las Palmas de Gran Canaria, 35016 Las Palmas de Gran Canaria, Spain; 3CIBER de Enfermedades Respiratorias, Instituto de Salud Carlos III, 28029 Madrid, Spain; jesus.villar54@gmail.com; 4Research Unit, Hospital Universitario de Gran Canaria “Doctor Negrín”, 35010 Las Palmas de Gran Canaria, Spain; 5Keenan Research Center for Biomedical Sciences at the Li-Ka Shing Knowledge Institute, St. Michael’s Hospital, Toronto, ON M5B 1W8, Canada

**Keywords:** active warming, body temperature, hypothermia, laparoscopic surgery, perioperative complications, pre-warming, urology

## Abstract

Background: Pre-warming prevents perioperative hypothermia. We evaluated the current clinical practice of pre-warming and its effects on temperature drop and postoperative complications; Methods: This prospective, observational pilot study examines clinical practice in a tertiary hospital on 99 patients undergoing laparoscopic urological surgery. Pre-warming was performed in the pre-anesthesia room. Patients were classified into three groups: P 0 (non-prewarmed), P 5–15 (pre-warming 5–15 min) and P > 15 (pre-warming 15–30 min). Tympanic temperature was recorded in the pre-anesthesia room, prior to anesthesia induction, and in the PACU. Esophageal temperature was recorded intraoperatively. The occurrence of shivering, pain intensity, length of stay in PACU, and postoperative complications during hospital stay were also recorded; Results: After pre-warming, between-group difference in body temperature was higher in P > 15 than in P 0 (0.4 °C, 95% CI 0.14–0.69, *p* = 0.004). Between P 5–15 and P 0 difference was 0.2 °C (95% CI 0.04–0.55, *p* = 0.093). Temperature at the end of surgery was higher in pre-warmed groups [mean between-group difference 0.5 °C (95% CI 0.13–0.81, *p* = 0.007) for P 5–15; 0.9 °C (95% CI 0.55–1.19, *p* < 0.001) for P > 15]. Pain and shivering was less common in pre-warmed groups. Postoperative transfusions and surgical site infections were lower in P > 15; Conclusion: Short-term pre-warming prior to laparoscopic urological surgery decreased temperature perioperative drop and postoperative complications.

## 1. Introduction

The laparoscopic approach for urological abdominal surgery decreases serious complications, such as surgical stress [1], perioperative bleeding [2], surgical wound infection [3], and hospital stay [4]. However, it has not been reported whether it could also decrease the rate of perioperative hypothermia when compared to an open approach [5]. Inadvertent hypothermia is common in the perioperative period, affecting 50–90% of patients undergoing surgery [6,7,8,9]. Perioperative hypothermia can increase intraoperative bleeding [10], surgical wound infection [11], discomfort, and length of hospital stay [12]. Thus, hypothermia throughout laparoscopic surgery can offset the expected benefits of this technique, and prevention of temperature drop during laparoscopic surgery should be mandatory.

Preoperative forced-air warming (pre-warming) is the most effective tool for preventing hypothermia. Its use is recommended in high-risk patients undergoing surgeries longer than 30 min [13,14,15,16,17], and it has been incorporated into routine clinical practice. However, trials examining the impact of pre-warming for preventing complications associated with hypothermia show conflicting results [18]. Despite the fact that laparoscopic surgeries are widespread nowadays, much is still unknown about the effect of pre-warming on these surgeries due to the lack of studies. In patients at risk for hypothermia undergoing laparoscopic surgery, pre-warming has been shown to prevent temperature drop [19]. However, the duration of prewarming and its effect on reducing complications have not yet been clarified.

We evaluated the effects of different time-periods of pre-warming on perioperative body temperature in patients undergoing urological laparoscopic surgery. As secondary objectives, we evaluated the relationship between a lower core temperature at the end of surgery and perioperative risk factors. We also tested the impact of pre-warming on the rate of postoperative shivering, pain, length of stay in Post-Anesthesia Care Unit (PACU), complications, and evolution of the patient.

## 2. Materials and Methods

This pragmatic, nonrandomized, prospective study evaluated routine clinical practice in 99 consecutive male patients scheduled to undergo laparoscopic prostatectomy or nephrectomy in a university hospital from August 2018 to October 2019. The study was conducted in accordance with the Declaration of Helsinki, and the protocol was approved by the Ethics Committee of Hospital Universitario de Gran Canaria Dr. Negrín, Las Palmas de Gran Canaria, Spain (IRB approval #2018-089-1), and prospectively registered at ClinicalTrials.gov (NCT03617809). All methods were carried out following good clinical practice. The manuscript follows the STROBE guidelines [20]. Patients were excluded if they met any of the following criteria: active infection, antipyretic consumption during the hospital stay, neuropathy, hyper or hypothyroidism, peripheral vascular disease, skin lesion, immunosuppression, insulin-dependent or poorly controlled diabetes mellitus, bleeding disorders, or intake of antiplatelet or anticoagulation agents. Written informed consent from each patient was obtained before enrollment.

### 2.1. Outcomes

For assessing the effect of different time-periods of pre-warming on perioperative body temperature, we selected as the primary outcome the difference in temperature between groups throughout the perioperative period and the first postoperative hour. Secondary objectives included the relationship between temperature at the end of surgery and perioperative risk factors, and the impact of pre-warming on the postoperative evolution of the patient.

### 2.2. Study Protocol

Upon hospital admission, the patient’s age, weight, height, American Society of Anesthesiologists (ASA) physical status, hemoglobin, and surgery (prostatectomy or nephrectomy) were recorded. Body mass index (BMI), body surface area (BSA), and basal metabolic rate (BMR) were calculated. Core temperature was measured using a tympanic thermometer (Genius-2 Tympanic Thermometer and Base, Covidien Ltd., Mansfield, MA, USA) at hospital admission and upon arrival at the pre-anesthesia room (PreT). Prior to initiating the study, nurses-in-charge of temperature monitoring were trained for temperature measurements [9]. To reduce intra-observer variability, the mean value of three consecutive measurements was selected.

Patients were pre-warmed at the pre-anesthesia room following routine clinical practice covering the entire body with a forced-air blanket (WarmTouch total body blanket, Covidien). The temperature output of the warmer (WarmTouch Model WT-5900, Covidien) was set at maximum level (43 °C). The duration of pre-warming depended on the time the patient had to wait in the pre-anesthesia room before surgery. Patients with a waiting time <5 min were not pre-warmed (P 0 group). Patients pre-warmed for 5–15 min were included in the P 5–15 group, and those pre-warmed for 15–30 min were included in the P > 15 group. Anesthesiologists responsible for clinical intraoperative management made no decisions on the duration of pre-warming and were not aware of its duration.

After pre-warming, the tympanic temperature was measured (T0). Then, patients were pre-medicated (intravenous midazolam, 1–2 mg) and monitored using non-invasive arterial pressure, electrocardiogram, peripheral oxygen saturation, and bispectral index (BIS, Covidien). General anesthesia was performed using remifentanil and propofol with effect-site target-controlled infusion (TCI, B.Braun, Melsungen, Germany) to maintain BIS 40–60 and intraoperative hemodynamic stability. Cisatracurium was administered in bolus (0.2 mg·kg^−1^) to allow orotracheal intubation and in continuous infusion during surgery to ensure a low intraperitoneal pressure during laparoscopy. Patients were actively warmed intraoperatively using a forced-air blanket over the upper part of the body. Temperature was measured intraoperatively using an esophageal thermometer (Mon-a-Therm, Covidien). Intraoperative core temperature was recorded at 15-min intervals (T15, T30, T45) during the first hour of anesthesia (T60), then at 30-min intervals (T90, T120, T150, T180, T210, T240, T270, T300) until the end of surgery (EndT). Intravascular fluids were not warmed. CO2 used for pneumoperitoneum was not heated or humidified. Volume of administered intravenous fluids, and intraoperative bleeding were registered.

At the end of surgery, the neuromuscular blockade was reversed and the tracheal tube was removed. Patients were transferred to the PACU, where an independent clinician was in-charge. Postoperative temperature during the first hour using tympanic thermometer, presence of shivering at arrival (using a dichotomous scale, positive when it was visible), pain intensity at 30 min after arrival (using a numerical rating scale, NRS), and length of stay in PACU were logged. Patients were transferred to the hospital ward once core temperature was above 36.0 °C and Aldrete modified score [21] was higher than 8. On the first postoperative morning, we measured hemoglobin for calculating the difference with the preoperative value. During postoperative hospital stay, blood transfusion requirements (as decided by an independent clinician) and presence of postoperative complications were recorded. Surgical site infection and postoperative ileus were clinically defined, assessed by an independent clinician. Follow-up ended once the patient was discharged from the hospital.

### 2.3. Statistical Analysis

The sample size was estimated based on a previous observational study [9], with a power analysis to detect a difference of 0.3 °C in core temperature. Ninety patients (*n* = 30 in each group) provided an 80% power for detecting a difference at an alpha-level of 0.05.

Data were analyzed using SPSS 24.0 (Statistical Package for Social Sciences, IBM). Data on categorical variables are expressed as frequency and percentage. A Chi-square test was used to compare frequency data among groups. Quantitative variables are expressed as mean ± SD. We used Shapiro–Wilk’s test to analyze the normality of data. To compare quantitative variables between the two groups, a *t*-test for independent samples was used in cases of variables with normal distribution, and Mann–Whitney *U*-test when the distribution of variables could not be adjusted to normality. To compare temperature (continuously scaled variable) among groups, one-way analysis of variance (ANOVA) test for independent samples was used in variables with normal distribution, and Kruskal-Wallis test where distribution was not adjusted to normality. Pearson’s correlation coefficient was used to detect relationships between quantitative perioperative variables and temperature at the end of surgery. Differences among groups regarding pain intensity, length of stay in PACU, postoperative hemoglobin, and hospital length of stay were analyzed using one-way ANOVA test for independent samples. A Chi-square test was used to compare the frequency of shivering and postoperative complications among groups. A *p*-value < 0.05 was considered statistically significant.

## 3. Results

Ninety-nine patients were included (*n* = 33 patients in each group). Patient characteristics, body temperature upon arrival at pre-anesthesia room (PreT), and perioperative characteristics were similar among groups (Table 1). After pre-warming, core temperature before entering the operating room (T0) was higher in P > 15 than in P 0 (between-groups difference 0.4 °C, 95% CI 0.14–0.69, *p* = 0.004). Between P 5–15—P 0 difference was 0.2 °C (95% CI 0.04–0.55, *p* = 0.093). Pre-warmed patients had a significantly higher temperature throughout the intraoperative period compared to non-pre-warmed patients (Table 2, Figure 1). The temperature at the end of the procedure (EndT) in P 0 was 35.8 ± 0.8 °C. EndT was higher in pre-warmed groups (between P 5–15—P 0 difference 0.5 °C, 95% CI 0.13–0.81, *p* = 0.007; between P > 15—P 0 difference 0.9 °C, 95% CI 0.55–1.19, *p* < 0.001). No secondary effects due to pre-warming, such as sweating or thermal discomfort were observed.

We found a direct proportional relationship among temperature and BSA, BMR, preoperative hemoglobin, operating room temperature, length of anesthesia, and temperature upon arrival at the pre-anesthesia room. We also found an inverse proportional relationship between EndT and volume of intravenous fluid therapy and intraoperative bleeding (Table 3).

Hypothermia upon the arrival at PACU occurred in 48.5% (16/33) of patients in P 0, 33% (11/33) in P 5–15 (RR 1.88, 95%CI 0.69–5.09, *p* = 0.211), and 6% (2/33) in P > 15 (RR 14.58, 95%CI 2.99–71.15, *p* < 0.001). Figure 2 shows the postoperative evolution of temperature throughout the first postoperative hour in PACU. Moreover, pre-warmed patients suffered from less postoperative pain and shivering (Table 4).

The estimated intraoperative bleeding was 377 ± 301 mL in P 0, 325 ± 257 mL in P 5–15, and 264 ± 189 mL in P > 15 (*p* = 0.204). No differences were found in postoperative hemoglobin or in the decrease in hemoglobin among groups (Table 4). However, we found statistically significant differences regarding the percentage of patients receiving postoperative transfusions: 18.2% (6/33) in P 0, 6.1% (2/33) in P 5–15 (RR 0.29, 95% CI 0.05–1.56, *p* = 0.131), and 0% in P > 15 (*p* = 0.010). Surgical site infection occurred in 12.1% (4/33) in P 0, 9.1% (3/33) in P 5–15 (RR 0.73 95% CI 0.15–3.53, *p* = 0.689), and 0% in P > 15 (*p* = 0.039). Postoperative ileus developed during the postoperative period in 15.2% (5/33) in P 0, and 9.1% (3/33) in both pre-warmed groups (RR 0.56, 95% CI 0.12–2.56, *p* = 0.451). No differences were found regarding PACU or hospital length of stay (Table 4).

## 4. Discussion

In this trial examining routine clinical practice regarding pre-warming, short-time periods of pre-warming (P 5–15 or P > 15) prior to urological laparoscopic surgery was able to increase core temperature intraoperatively and during the first postoperative hour. These short-time periods of pre-warming decreased the rate of hypothermia, the occurrence of shivering, the intensity of pain, and the need for transfusions postoperatively.

It could be argued that there is a low risk of hypothermia during laparoscopic surgery because no heat loss occurs from exposure of surgical wounds and abdominal organs to the environment. However, in our study, up to 29.3% of patients developed hypothermia due to internal heat redistribution, despite preventive measures, and intraoperative hypothermia is also observed during laparoscopic procedures [22]. The mechanism underlying hypothermia in laparoscopic procedures has not been well elucidated. One of main reasons could be the prolonged surgical time and the increased heat loss via exposure to cold/dry CO_2_ insufflation during pneumoperitoneum [23]. In non-laparoscopic surgeries, pre-warming for less than an hour has been shown to decrease the occurrence of intraoperative hypothermia [9,24,25,26,27,28]. However, few studies have examined the effect of pre-warming in laparoscopic surgery. In a small randomized clinical trial in laparoscopic cholecystectomy, patients pre-warmed for one hour showed a lower rate of hypothermia during the first intraoperative hour [29]. However, pre-warming for an hour might be too long to be included in routine clinical practice.

Pre-warming prevents hypothermia by decreasing the temperature gradient between peripheral and central compartments, not by increasing core temperature [30]. However, in our study, pre-warming for 15–30 min increased the body temperature prior to anesthetic induction. During the first intraoperative hour, body temperature dropped sharply in all groups, as expected [31]. Afterward, the temperature remained constant until the end of surgery in non-pre-warmed patients, probably due to intraoperative active warming. Nevertheless, pre-warmed patients showed an exponential increase in temperature from the first intraoperative hour to the end of surgery and this effect persisted up to the first postoperative hour [32]. Therefore, pre-warming could enhance the effectiveness of intraoperative warming, increasing perioperative body temperature. In short-term surgeries, pre-warming diminishes the incidence of hypothermia in PACU and shortens length of stay [9]. Although pre-warming prevents against shivering and postoperative pain [9,33], we found no differences in the length of stay in PACU. Duration of recovery after long-term surgeries may depend on several factors, such as the need for postoperative care (control of drainage or diuresis), or the routine institutional protocol [27,34].

Hypothermia is known to increase perioperative bleeding by causing alterations in platelet and coagulation enzyme function [10,35,36], and intraoperative normothermia reduces perioperative bleeding [2,37]. However, these hematological alterations may not have a clinical impact on increasing intraoperative bleeding [38,39]. Improvement in surgical techniques and less invasiveness in laparoscopic surgery imply a decrease in intraoperative bleeding, making it more difficult to detect intergroup differences. Of note, we did not find a reduction in intraoperative bleeding or a lower drop in hemoglobin values in pre-warmed patients, but we found a decrease in the requirement of postoperative transfusions in patients pre-warmed for 15–30 min. Hypothermia also causes an impairment in immune function and vasoconstriction, increasing the risk of surgical wound infection [12]. In our study, pre-warming for 15–30 min reduced the incidence of surgical wound infection, as previously demonstrated [40]. However, pre-warming for 5–15 min was not efficiently enough to reach statistical significance, likely because core temperature in P 5–15 group was lower than in the P > 15 group. On the other hand, hypothermia is not an independent risk factor for postoperative ileus, but its appearance delays the onset of oral tolerance [41]. We found no influence of pre-warming on decreasing the rate of postoperative ileus, probably due to its low prevalence in laparoscopic urological surgery.

We acknowledge some potential limitations and strengths. First, since this is an observational study, we can only report associations. Patients were not randomized into different pre-warmed groups, and pre-warming time was selected arbitrarily. However, its design faithfully reflects the true relevance of short pre-warming on routine clinical practice in laparoscopic urological surgery. Secondly, since the study only included males, results cannot be generalized to the female population. Third, we used non-invasive tympanic and esophageal thermometers to monitor temperature depending on the moment of measurement (tympanic when the patient was awake, esophageal intraoperatively). However, we believe that our results are valid since comparison among groups at each measurement was performed using the same thermometer.

## 5. Conclusions

Pre-warming for 5–15 min and 15–30 min prior to laparoscopic urological surgery reduces the occurrence of hypothermia during the intraoperative period and during the first postoperative hour. Pre-warming for 15–30 min increased core temperature before induction and reduced occurrence of hypothermia upon arrival at PACU. Both short-time periods of pre-warming decreased the rate of shivering and the intensity of pain in PACU. Pre-warming for 15–30 min reduced the need for postoperative transfusions and the prevalence of surgical site infections. Thus, the take-home message is that less than 5 min of pre-warming is useless and more than 15 min is better than less.

## Figures and Tables

**Figure 1 jcm-10-01047-f001:**
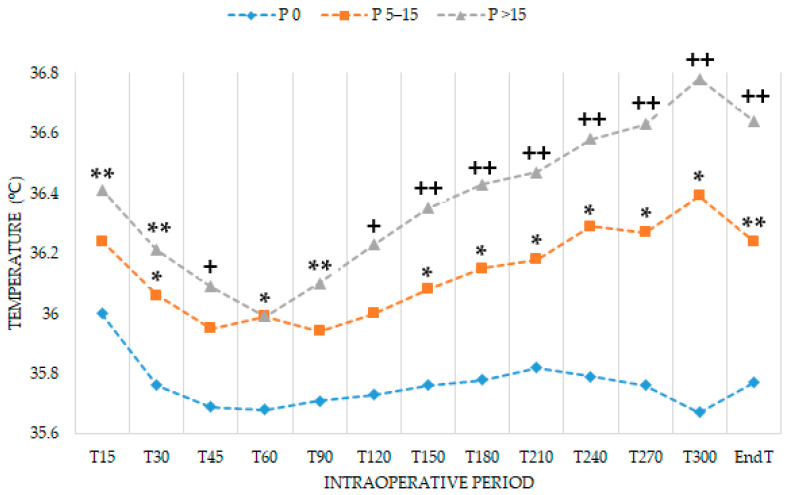
Mean perioperative temperature (°C) in each group. PreT: temperature upon arrival at the pre-anesthesia room; EndT: temperature at the end of surgery; P 0: non-pre-warmed; P 5–15: pre-warmed for 5–15 min; P > 15: pre-warmed for 15–30 min. * *p* < 0.05 vs. P 0; ** *p* < 0.01 vs. P 0; + *p* < 0.001 vs. P 0; ++ *p* ≤ 0.0001 vs. P 0.

**Figure 2 jcm-10-01047-f002:**
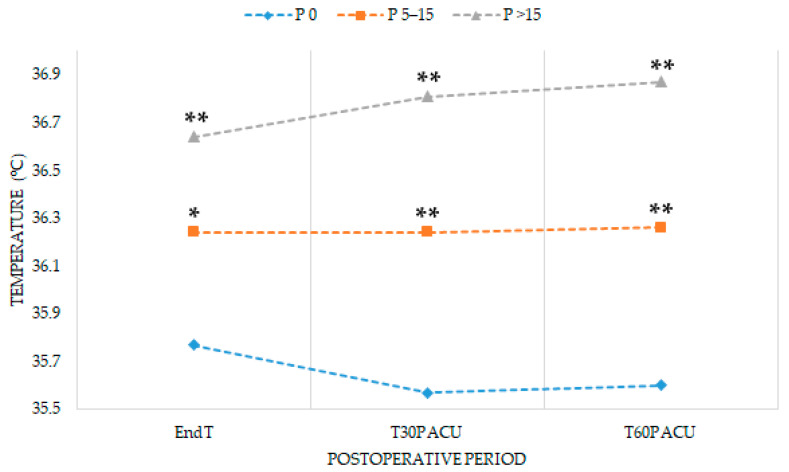
Mean perioperative temperature (°C) in the PACU in each group. PACU: Post-Anesthetic Care Unit; EndT: temperature at the end of surgery; T30PACU: temperature 30 min after admission to PACU; T60PACU: temperature 60 min after admission to PACU; P 0: non-pre-warmed; P 5–15: pre-warmed for 5–15 min; P > 15: pre-warmed for 15–30 min. * *p* < 0.01 vs. P 0; ** *p* < 0.0001 vs. P 0.

**Table 1 jcm-10-01047-t001:** Patient characteristics and perioperative variables.

	P 0 (*n* = 33)	P 5–15 (*n* = 33)	P > 15 (*n* = 33)	*p*
Age (years)	59.7 ± 9.2	59.2 ± 12.2	60.5 ± 8.1	0.895
BMI (kg·m^−2^)	27.9 ± 5.6	27.2 ± 3.7	28.2 ± 3.5	0.581
BSA (m^2^)	1.9 ± 0.3	1.9 ± 0.2	1.9 ± 0.2	0.451
BMR (kcal)	1641 ± 273	1587 ± 232	1667 ± 203	0.516
ASA II (%)	60.6	57.6	60.6	0.959
Core temperature at admission (°C)	36.5 ± 0.5	36.4 ± 0.5	36.4 ± 0.5	0.274
Hypothermia (<36 °C) upon arrival at pre-anesthesia room (%)	21.2	36.4	12.1	0.062
Laparoscopic Surgery (Prostatectomy, %)	57.6	51.5	66. 7	0.399
Hemoglobin at admission (g·dL^−1^)	14.3 ± 2.1	14.3 ± 2.2	14.5 ± 1.9	0.890
Duration of anesthesia (min)	280 ± 60	291 ± 66	302 ± 57	0.444
Operating room temperature (°C)	22.6 ± 0. 6	22.6 ± 0.6	22.7 ± 0.6	0.714
Fluid therapy (mL)	2042 ± 1070	1795 ± 688	1759 ± 563	0.654

Data are expressed as mean ± SD and percentage. BMI: body mass index; BSA: body surface area; BMR: basal metabolic rate; ASA: American Society of Anesthesiologists; P 0: non-pre-warmed; P 5–15: pre-warmed for 5–15 min; P > 15: pre-warmed for 15–30 min.

**Table 2 jcm-10-01047-t002:** Perioperative core temperature evolution.

	P 0 (*n* = 33)	P 5–15 (*n* = 33)	P > 15 (*n* = 33)	*p*
PreT	36.3 ± 0.6	36.2 ± 0.5	36.3 ± 0.4	0.73
T0	36.2 ± 0.7	36.5 ± 0.5	36.6 ± 0.4	0.009
T15	36.0 ± 0.6	36.2 ± 0.4	36.4 ± 0.4	0.003
T30	35.8 ± 0.6	36.1 ± 0.5	36.2 ± 0.4	0.002
T45	35.7 ± 0.6	35.9 ± 0.5	36.1 ± 0.4	0.007
T60	35.7 ± 0.6	35.9 ± 0.4	35.9 ± 0.4	0.014
T90	35.7 ± 0.6	35.9 ± 0.5	36.1 ± 0.5	0.009
T120	35.7 ± 0.6	36.0 ± 0.5	36.2 ± 0.5	0.001
T150	35.8 ± 0.6	36.1 ± 0.5	36.3 ± 0.5	<0.001
T180	35.8 ± 0.7	36.2 ± 0.5	36.4 ± 0.4	<0.001
T210	35.8 ± 0.8	36.2 ± 0.5	36.5 ± 0.5	<0.001
T240	35.8 ± 0.8	36.3 ± 0.5	36.6 ± 0.5	<0.001
T270	35.8 ± 0.8	36.3 ± 0.6	36.6 ± 0.5	0.0001
T300	35.7 ± 0.9	36.4 ± 0.5	36.8 ± 0.5	0.0001
EndT	35.8 ± 0.8	36.2 ± 0.6	36.6 ± 0.5	<0.0001

Data are expressed as mean ± SD. PreT: temperature upon arrival at the pre-anesthesia room; T0: temperature after pre-warming; T15: temperature 15 min after induction; T30: temperature 30 min after induction; T45: temperature 45 min after induction; T60: temperature 60 min after induction; T90: temperature 90 min after induction; T120: temperature 120 min after induction; T150: temperature 150 min after induction; T180: temperature 180 min after induction; T210: temperature 210 min after induction; T240: temperature 240 min after induction; T270: temperature 270 min after induction; T300: temperature 300 min after induction; EndT: temperature at the end of surgery; P 0: non-pre-warmed; P 5–15: pre-warmed for 5–15 min; P > 15: pre-warmed for 15–30 min.

**Table 3 jcm-10-01047-t003:** Correlation between EndT and characteristics of patients and perioperative variables.

	Age	BMI	BSA	BMR	Pre Hb	OR Temp	Length Anesthesia	Fluid Therapy	PreT	Bleeding
Pearson correlation	−0.19	0.14	0.22	0.24	0.28	0.32	0.27	−0.31	0.24	−0.22
*p*	0.065	0.162	0.030	0.018	0.005	0.001	0.006	0.002	0.018	0.031

BMI: body mass index; BSA: body surface area; BMR: basal metabolic rate; Pre Hb: preoperative hemoglobin; OR Temp: operating room temperature; PreT: core temperature upon arrival at the pre-anesthesia room.

**Table 4 jcm-10-01047-t004:** Postoperative evolution of patients in PACU, postoperative lab test, and complications throughout hospital stay and length of hospital stay.

	P 0 (*n* = 33)	P 5–15 (*n* = 33)	P > 15 (*n* = 33)	*p*
NRS in PACU	3.06 ± 2.2	1.79 ± 1.2 *	1.58 ± 1.3 **	0.001
Shivering in PACU (%)	39.39	3.03	3.03	<0.001
Length of stay in PACU (min)	531 ± 443	482 ± 375	449 ± 331	0.69
Postoperative Hemoglobin (g·dL^−1^)	11.36 ± 1.6	11.73 ± 1.9	11.84 ± 1.8	0.52
Hemoglobin drop (g·dL^−1^)	2.93 ± 1.4	2.60 ± 1.7	2.67 ± 1.7	0.67
Hospital length of stay (days)	4.5 ± 3	3.7 ± 3	3.4 ± 2.4	0.24

Data are expressed as mean ± SD and percentage. NRS: numerical rating scale; PACU: Post-Anesthetic Care Unit; P 0: non-pre-warmed; P 5–15: pre-warmed for 5–15 min; P > 15: pre-warmed for 15–30 min. * *p* = 0.005 vs. P 0; ** *p* = 0.001 vs. P 0.

## Data Availability

Data presented in this study are available on request from the corresponding author.

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
