# Peer review of "Short-Periods of Pre-Warming in Laparoscopic Surgery. A Non-Randomized Clinical Trial Evaluating Current Clinical Practice"

_jcm, 2021, doi:10.3390/jcm10051047_

Round 1
Reviewer 1 Report
We consider that there are 2 major methodoligical concerns. First, the authors separated the patients into 3 groups according to the prewarming time: less than 5 minutes, between 5 and 15 minutes and more than 15 minutes. On which basis this time limits have been statued? Also, it's surprising to consider that a patient with 16 minutes of prewarming belongs to the same group than a patient prewarmed during 29 minutes. Search of a logistic regression between Tc at different times and duration of prewarming might be more interesting.
The second point is the use of 2 different thermometers with 2 different technologies on 2 different sites. Combination of the 2 different measurements on Figure 1 is probably inappropriate and mean temperatures displayed on the graph should be limited to temperatures measured with the esophageal probe.
A surprising point is that core temperature significantly increases during prewarming in awake patients with a warmer set at 43°C. The body heat content is thermoregulated. An increase in skin temperature induces a decrease in core temperature and triggers sweating. The authors don't report sweating or thermal discomfort. We can observe it if the patient is more than 70 years old or in case of administration of a medication altering thermoregulatory thresholds (i.e. midazolam). The authors should clarify this point.
That's an important point since the benefit of prewarming in the study corresponds to the initial increase in core temperature. After the induction of the anesthesia the mean drop in Tc (0.5 to 0.6°C) is similar in all the groups.
It's also surprising to observe that the internal redistribution range is not modified by the prewarming duration whereas the thermal gradient is sharply reduced.
Comments
Introduction section:
The objective of the study reported at the end of the introduction must be corrected. It suggests that there are 2 groups clearly defined (15 or 30 min). It doesn't correspond to the design of the study.
Discussion section:
Lines 205 to 218: laparoscopic surgery probably reduces heat losses comparatively to open surgery. But, the initial core temperature decrease occuring during the 1st hour is mainly related to the internal redistribution which is not modified by the surgical technique.
Paragraph 4 (lines 234 to 250) could be deleted. Transfusion or bleeding are not only related to intraoperative core temperature management.
Author Response
Response to Reviewer 1 Comments:
Point 1) We consider that there are 2 major methodological concerns. First, the authors separated the patients into 3 groups according to the prewarming time: less than 5 minutes, between 5 and 15 minutes and more than 15 minutes. On which basis this time limits have been stated? Also, it's surprising to consider that a patient with 16 minutes of prewarming belongs to the same group than a patient prewarmed during 29 minutes. Search of a logistic regression between Tc at different times and duration of prewarming might be more interesting.
Response 1- Thank you for this comment. It would have been interesting to treat pre-warming duration as a continuous variable and compare it with body temperature at different times intraoperatively. However, we designed our study based on a previous report by our group in which the duration of pre-warming was stratified (Becerra et al, Sci Rep 2019, 9:16477), and the aim was to stablish pragmatically the effect of different pre-set times of pre-warming on body temperature. We considered that this would increase the applicability of the inferred data. In addition, by grouping the different pre-warming periods, we could achieve a greater number of patients in each group and an adequate statistical power.
Point 2) The second point is the use of 2 different thermometers with 2 different technologies on 2 different sites. Combination of the 2 different measurements on Figure 1 is probably inappropriate and mean temperatures displayed on the graph should be limited to temperatures measured with the esophageal probe.
Response 2- Thank you for the opportunity to clarify this issue. We agree with the Reviewer that the use of two different technologies for measuring the body temperature is a limitation of our study, and as such, we included this observation as a limitation in the Discussion section (lines 269 – 271). However, comparison among groups was performed using the same thermometer (tympanic when the patient was awake and esophageal when the patients was under anesthesia), and our results should be considered valid, although combining values obtained by two different methods might be seen as a limitation. Following the Reviewer’s recommendation, Figure 1 has been now changed accordingly in the revised manuscript, displaying on the graph only temperatures measured intraoperatively with the esophageal thermometer.
Point 3) A surprising point is that core temperature significantly increases during prewarming in awake patients with a warmer set at 43°C. The body heat content is thermoregulated. An increase in skin temperature induces a decrease in core temperature and triggers sweating. The authors don't report sweating or thermal discomfort. We can observe it if the patient is more than 70 years old or in case of administration of a medication altering thermoregulatory thresholds (i.e. midazolam). The authors should clarify this point.
Response 3- Thanks for this comment. We would like to confirm to the Reviewer that no secondary effects due to pre-warming, such as sweating or thermal discomfort were observed in our patient population. We have added this information in the Results section of the revised manuscript (lines 157 – 158). We would like to clarify that the application of pre-warming was performed in the pre-anesthesia room, prior to the administration of any medication that may alter the thermoregulation of patients. Therefore, we consider that there could be no interference between the medication and the absence of secondary effects due to pre-warming.
Point 4) That's an important point since the benefit of prewarming in the study corresponds to the initial increase in core temperature. After the induction of the anesthesia the mean drop in Tc (0.5 to 0.6°C) is similar in all the groups. It's also surprising to observe that the internal redistribution range is not modified by the prewarming duration whereas the thermal gradient is sharply reduced.
Response 4- We deeply thank the Reviewer for this positive comment about our study. In fact, we believe that one of the strengths of this pragmatic study is that body temperature drops in all groups similarly during the first intraoperative hour, as it has been widely described in the literature. However, temperature of pre-warmed groups increases rapidly from the first hour, while in the control group temperature remains stable during the rest of the intervention (as we have stated in the Discussion section of the manuscript, lines 233 – 239).
Point 5) Comments: Introduction section: The objective of the study reported at the end of the introduction must be corrected. It suggests that there are 2 groups clearly defined (15 or 30 min). It doesn't correspond to the design of the study.
Response 5- Thank you. Following the Reviewer’s suggestion, we have corrected the wording of the objective of the study in the Introduction section of the revised manuscript. We have now written that “we evaluated the effects of different time-periods of pre-warming on perioperative body temperature in patients undergoing urological laparoscopic surgery.” (lines 53 – 54).
Point 6) Discussion section: Lines 205 to 218: laparoscopic surgery probably reduces heat losses comparatively to open surgery. But, the initial core temperature decrease occurring during the 1st hour is mainly related to the internal redistribution which is not modified by the surgical technique.
Response 6- Thank you very much for this comment. As noticed, the initial temperature drop is related to the internal redistribution (we have added explicitly this information in line 221). However, in our study we also showed the impact of pre-warming in reducing the temperature drop throughout the entire intraoperative period and this effect persisted up to the first postoperative hour. We aimed to summarize the possible mechanisms underlying the presence of hypothermia in laparoscopic surgery (lines 222 – 225) and to extrapolate data obtained from pre-warming in open approach to laparoscopic approach (lines 225 – 227).
Point 7) Paragraph 4 (lines 234 to 250) could be deleted. Transfusion or bleeding are not only related to intraoperative core temperature management.
Response 7- We agree with the Reviewer that bleeding is not related to intraoperative core temperature management (as pointed out in lines 247 – 250). However, since we recorded the impact of pre-warming on common postoperative complications, we considered postoperative bleeding as useful information for the reader and compare it with other studies regarding the impact of body temperature in perioperative bleeding.
Reviewer 2 Report
The authors evaluated the effects of different time-periods of pre-warming (15 or 30 min) on perioperative body temperature in patients undergoing urological laparoscopic surgery. They also evaluated the relationship between a lower core temperature at the end of surgery and perioperative risk factors. They concluded that short-term pre-warming prior to laparoscopic urological surgery decreased temperature perioperative drop and postoperative complications.
I read the study with great interest. The study is well designed, with adequate methodology and results interpretation, followed by very good discussion. I really congratulate to the authors on this interesting, well designed and nicely written study. I do not have major remarks except that the authors should randomize the patients, this would significantly increase the validity of the study, but the authors correctly listed this as limitation of the study.
Minor remarks:
- Abstract – Please remove numbers 1 – 4 in squares prior to subtitles.
- Introduction – The authors stated that: ''The laparoscopic approach for urological abdominal surgery decreases serious complications, such as perioperative bleeding [1], surgical wound infection [2], and hospital stay [3]''. I would like to point importance of surgical stress as important factor which is reduced in laparoscopic surgery, compared to open surgery. I would advise to the authors to include ‘’reduced surgical stress’’ in this sentence together with reference: Jukić M, et al. Comparison of inflammatory stress response between laparoscopic and open approach for pediatric inguinal hernia repair in children. Surg Endosc. 2019;33(10):3243-3250. doi: 10.1007/s00464-018-06611-y.
- Please provide primary and secondary outcomes of the study in methodology (not in introduction).
- Table 2. – Please include explanation of T0 – T15 – T30 – T 45 - …. T 300 in the legend of the Table.
- Table 2 – All values should be presented as ±, not +. Please revise!
Author Response
Response to Reviewer 2 Comments:
Point 1) The authors evaluated the effects of different time-periods of pre-warming (15 or 30 min) on perioperative body temperature in patients undergoing urological laparoscopic surgery. They also evaluated the relationship between a lower core temperature at the end of surgery and perioperative risk factors. They concluded that short-term pre-warming prior to laparoscopic urological surgery decreased temperature perioperative drop and postoperative complications.
I read the study with great interest. The study is well designed, with adequate methodology and results interpretation, followed by very good discussion. I really congratulate to the authors on this interesting, well designed and nicely written study. I do not have major remarks except that the authors should randomize the patients, this would significantly increase the validity of the study, but the authors correctly listed this as limitation of the study.
Response 1- We thank the Reviewer for the positive comments about our study.
Point 2) Abstract – Please remove numbers 1 – 4 in squares prior to subtitles.
Response 2- Thank you. Done: Numbers have been removed from the Abstract.
Point 3) Introduction – The authors stated that: ''The laparoscopic approach for urological abdominal surgery decreases serious complications, such as perioperative bleeding [1], surgical wound infection [2], and hospital stay [3]''. I would like to point importance of surgical stress as important factor which is reduced in laparoscopic surgery, compared to open surgery. I would advise to the authors to include ‘’reduced surgical stress’’ in this sentence together with reference: Jukić M, et al. Comparison of inflammatory stress response between laparoscopic and open approach for pediatric inguinal hernia repair in children. Surg Endosc. 2019;33(10):3243-3250. doi: 10.1007/s00464-018-06611-y.
Response 3- We thank the Reviewer for this comment. We have now added in the Introduction section of the revised manuscript the reduction of surgical stress as one of the advantages of the laparoscopic approach (line 34). Since we have added the suggested new reference, the rest of references throughout the manuscript have changed their number.
Point 4) Please provide primary and secondary outcomes of the study in methodology (not in introduction).
Response 4- Thank you. Primary and secondary outcomes of the study have been added in a subsection in the methodology section of the revised manuscript as follows: “For assessing the effect of different time-periods of pre-warming on perioperative body temperature, we selected as the primary outcome the difference in temperature between groups throughout the intraoperative period and the first postoperative hour. Secondary objectives included the relationship between temperature at the end of surgery and perioperative risk factors, and the impact of pre-warming on the postoperative evolution of the patient.” (lines 73 – 79).
Point 5) Table 2. – Please include explanation of T0 – T15 – T30 – T 45 - …. T 300 in the legend of the Table.
Response 5- Thank you. Following the Reviewer recommendation, we have added an explanation for the intraoperative time of temperature measurement in the legend of Table 2 (lines 164 – 170).
Point 6) Table 2 – All values should be presented as ±, not +. Please revise!
Response 6- Thank you. All tables have been revised.
Round 2
Reviewer 1 Report
I'm still concerned by the way to classify the patients into 3 groups suggesting that the time of prewarming was 0 or exactly 15 min or 30 min.
In fact it could be more appropriate to entitle the groups : 0, Between 5 and 15 min, and >15min.
The "take home messages" could be easily drawn : less than 5 min of prewarming is useless and more than 15 min is better than less.
Author Response
Response to Reviewer 1 Comments:
Point 1) I’m still concerned by the way to classify the patients into 3 groups suggesting that the time of prewarming was 0 or exactly 15 min or 30 min.
In fact it could be more appropriate to entitle the groups: 0, Between 5 and 15 min, and >15min.
Response 1- Thank you for this comment. Throughout the entire revised manuscript, groups have been re-entitled as suggested by the Reviewer: P0, P5-15, and P>15.
Point 2) The “take home messages” could be easily drawn: less than 5 min of prewarming is useless and more than 15 min is better than less.
Response 2- Thank you. We have added this sentence in the conclusion paragraph of the manuscript (lines 281 – 282).